# Outcomes of first emergency admissions for alcohol-related liver disease in England over a 10-year period: retrospective observational cohort study using linked electronic databases

Keith Bodger  ,[1,2] Thomas Mair,[1] Peità Schofield,[3] Benjamin Silberberg,[1] Steve Hood,[2] Kate M Fleming[4]

[1]Department of Health Data Science, Institute of Population Health, University of Liverpool, Liverpool, UK
[2]Gastroenterology Department, Liverpool University Hospitals NHS Foundation Trust, Liverpool, UK
[3]Department of Public Health & Policy, University of Liverpool, Liverpool, UK
[4]Data & Analytics Transformation Directorate, NHS England, Redditch, UK

**Correspondence to**
Dr Keith Bodger;
kbodger@liverpool.ac.uk

## ABSTRACT

**Objectives** To examine time trends in patient characteristics, care processes and case fatality of first emergency admission for alcohol-related liver disease (ARLD) in England.

**Design** National population-based, retrospective observational cohort study.

**Setting** Clinical Practice Research Datalink population of England, 2008/2009 to 2017/2018. First emergency admissions were identified using the Liverpool ARLD algorithm. We applied survival analyses and binary logistic regression to study prognostic trends.

**Outcome measures** Patient characteristics; 'recent' General Practitioner (GP) consultations and hospital admissions (preceding year); higher level care; deaths in-hospital (including certified cause) and within 365 days. Covariates were age, sex, deprivation status, coding pattern, ARLD stage, non-liver comorbidity, coding for ascites and varices.

**Results** 17 575 first admissions (mean age: 53 years; 33% women; 32% from most deprived quintile). Almost half had codes suggesting advanced liver disease. In year before admission, only 47% of GP consulters had alcohol-related problems recorded; alcohol-specific diagnostic codes were absent in 24% of recent admission records. Overall, case fatality rate was 15% in-hospital and 34% at 1 year. Case-mix-adjusted odds of in-hospital death reduced by 6% per year (adjusted OR (aOR): 0.94; 95% CI: 0.93 to 0.96) and 4% per year at 365 days (aOR: 0.96; 95% CI: 0.95 to 0.97). Exploratory analyses suggested the possibility of regional inequalities in outcome.

**Conclusions** Despite improving prognosis of first admissions, we found missed opportunities for earlier recognition and intervention in primary and secondary care. In 2017/2018, one in seven were still dying during index admission, rising to one-third within a year. Nationwide efforts are needed to promote earlier detection and intervention, and to minimise avoidable mortality after first emergency presentation. Regional variation requires further investigation.

## INTRODUCTION

Alcohol-related liver disease (ARLD) is a leading cause of premature death and is

### STRENGTHS AND LIMITATIONS OF THIS STUDY

⇒ Nationwide population-based study sample covering all regions of England.
⇒ Application of a novel cohort discovery algorithm and the use of linked datasets covering primary and secondary care and death certification.
⇒ The datasets lack some important case-mix information such as physiological or biochemical measures of disease severity which would allow for further risk-adjustment of case fatality rates.
⇒ The cases are drawn from a large, representative sample of the English population but the findings may not be generalisable to all parts of the country.

set to overtake ischaemic heart disease for working years-of-life-lost in the UK.[1] Major shortcomings in acute hospital treatment were highlighted by the National Confidential Enquiry into Patient Outcome and Death (NCEPOD).[2] This widely publicised enquiry, combined with reports from the *Lancet Standing Commission* on Liver Disease,[3] prompted UK initiatives to promote better care.[4–8] Similar calls for action have been raised in other countries.[9] However, in their final report, the commission's authors highlighted continuing concerns about late diagnosis and unacceptably high levels of mortality in hospitalised patients.[10]

A patient's first emergency hospital admission is often the point of initial diagnosis.[11] High quality care is crucial to minimise early mortality and allow time for engagement of services promoting abstinence and recovery. However, knowledge of characteristics and outcomes of first admissions is limited. Nor is it known whether case-mix has changed or prognosis improved over time.

Hospital administrative data are the main source of national statistics[12] and underpin

proposed metrics.[13] However, identifying and characterising cases using routine data is challenging.[11] Patients present to hospital with varied clinical manifestations and co-existing alcohol-related conditions.[14] Analysis of administrative data in our region revealed that standard case definitions (based on primary discharge diagnoses alone)[12] identified only half of true admissions.[11] We developed an algorithm to identify combinations of primary and secondary codes compatible with acute presentations of ARLD.[11] Recent national testing confirmed a doubling of estimates of annual admissions using this method,[15] representing an under-recording of NHS hospital activity of £70–127 million for 2020.[15] This prompted calls for revising methods for national statistics.[16]

We undertook a retrospective cohort study using linked electronic datasets and the Liverpool ARLD algorithm (LAA).[11 15] We aimed to describe sociodemographics, clinical features, care processes (prior healthcare contacts in preceding year, length of stay and provision of higher level care) and case fatality of patients presenting with their first emergency admission for ARLD in England. We examined trends over a 10-year period, spanning 5 years before and after the NCEPOD report.[2] We also undertook exploratory analyses of regional variation in admission outcome.

## METHODS
### Data sources
The cohort was derived from the Clinical Practice Research Datalink (CPRD) containing linked data from primary and secondary care and death certification (Office for National Statistics).[17] This anonymised database covers a broadly representative sample of UK residents. The primary care consultation records contain diagnoses encoded as READ codes. Each case in the present study had linkage to Hospital Episode Statistics (HES), comprising administrative data for inpatient care delivered in NHS hospitals (available for the period April 1997 to March 2021). Care episodes in HES contain diagnostic codes generated after discharge using the international classification of diseases (ICD-10) system. The dataset also included the Adult Critical Care Minimum Dataset.

### Population of interest: first emergency admissions for ARLD
We extracted data for all adult patients (≥18 years) with a first unplanned (non-elective) hospital admission (spell) for ARLD. We defined a 'first' (or index) admission as having no such emergency admission within the preceding 10 years. We focused on prior emergency hospitalisations (ie, we did not screen elective admissions). We included all completed index emergency discharge records between 1 April 2008 and 31 March 2018, thereby covering 10 fiscal years (2008/2009 to 2017/2018). The cohort was restricted to residents with a record of CPRD practice registration ≥1 year.

### Case definitions for ARLD admissions
We used the LAA to identify admissions, requiring discharge codes to comply with one of two alternative case definitions ('primary' or 'uplift' subgroups).[11] The primary definition selects cases with a K70.x code as primary diagnosis, as in national statistics.[12] The 'uplift' definition identifies additional admissions with non-primary coding patterns[11 15] Briefly, uplift cases have a primary code for either a symptom, sign or complication of liver disease or for another alcohol-specific condition, accompanied by secondary codes compatible with ARLD (either K70.x or code for unspecified liver disease provided additional alcohol-specific conditions are listed).

### Stage of liver disease
We further characterised cases by assigning 'recorded-stage' of liver disease based on the K70.x code present in the index emergency admission (ie, alcoholic fatty liver (K70.0), hepatitis (K70.1), sclerosis and fibrosis (K70.2), cirrhosis (K70.3), hepatic failure (K70.4) or 'unspecified' (K70.9)). All primary and most uplift admission records contain a K70.x code. A minority of uplift cases lack a K70.x code, whereby a non-specific liver disease code (eg, 'Hepatic failure, unspecified' or 'Other and unspecified cirrhosis of liver') is accompanied by codes for other alcohol-specific conditions. In these cases, the relevant stage was taken from the liver code descriptor.

### Patient-level covariates derived from the index admission
Covariates were age, age group, sex, ethnicity, deprivation status, case definition (primary or uplift), recorded-stage of liver disease, coding for specific features of advanced liver disease (varices and ascites) and comorbidities. Socioeconomic deprivation was defined by the English indices of deprivation 2015 and grouped into quintiles (1=Least deprived, 5=Most deprived), as described previously.[11 18] Comorbidities listed in the Charlson Comorbidity Index[19] were identified but we excluded liver disease codes (mild and moderate-to-severe liver disease categegories) and dichotomised cases into two groups (score 0–1 and ≥2).

### Contacts in the year prior to index admission
The earliest stage of ARLD (fatty liver) may develop over months but more severe liver disease (hepatitis, cirrhosis or liver failure) requiring hospitalisation requires several years of harmful drinking.[14 20] We focused on contacts within a year before index emergency admission, assuming all such patients would have an established history of alcohol excess. The absence of relevant codes within such encounters would be taken as a marker of 'missed opportunities' to screen, detect and/or intervene during the year before liver-related emergency hospitalisation.

### Consultations in primary care
All available primary care contacts within the prior year were extracted and screened. For each contact, we created two binary variables. First, a variable to indicate whether

any alcohol-related code was present—signifying whether alcohol misuse or harms were suspected/recognised and recorded at the time. We used an extensive list of READ codes (see online supplemental appendix 1). Second, whether a diagnosis of liver disease was recorded—a marker of prior suspicion, recognition or elective diagnosis of ARLD.

### Emergency hospital admissions

We identified all episodes of unplanned hospital care in the year prior to index emergency admission and screened the diagnosis fields for any alcohol-specific ICD-10 code. Hence, we determined whether any non-liver related alcohol problems were diagnosed and recorded during those prior admissions.

### Care processes

We determined length of hospital stay. Using the Adult Critical Care Minimum Dataset, we determined whether the patient received higher level care during their index admission, including high dependency (category 2) and intensive care (category 3).

### Time periods of interest

We defined two 5-year time periods to compare cases discharged in the first and second half of the decade (period 1: April 2008 to March 2013; period 2: April 2013 to March 2018). We also created a variable to indicate fiscal year of discharge.

### Outcomes

We examined all-cause case fatality identified from death certification. We created binary variables for death during index emergency admission (in-hospital death) and tracked all cases for death within 365 days. For survival analysis we calculated time-to-death (days from admission). We used case fatality rates (CFRs), as opposed to population-based relative mortality measures, since our purpose was to generate metrics focused on effectiveness of acute care after adjusting for severity of presenting illness.

### Certified cause of in-hospital death

We accessed linked records of death certification for all in-hospital deaths during index admissions to identify deaths attributable to ARLD. This was based on screening all available 'cause' fields for descriptors relating to deaths attributable to liver disease and alcohol. See online supplemental appendix 2 for details.

### Statistical analyses

First, we summarised characteristics overall and for relevant strata and examined time trends. Categorical variables are reported as counts and percentages (groups compared using $\chi^2$ statistics). Continuous variables are reported as mean (SD), or median (IQR), as appropriate with comparisons made using parametric (t-test or analysis of variance test (ANOVA)) or non-parametric tests (Mann-Whitney U test or Kruskal-Wallis).

Second, we calculated crude CFRs as count of deaths (numerator) divided by number of first admissions (denominator) per time period. We report deaths in-hospital and at 365 days. We compared 5-year periods ($\chi^2$ test) and significance of monotonic year-on-year trends (Mann Kendall trend test). Kaplan-Meier survival analysis compared cumulative probability of death up to 365 days, using log-rank test to determine significance between strata. All cases had at least 1 year follow-up (no censoring).

Third, we determined factors associated with death using binary logistic regression. Models were used to adjust CFR for significant covariates. The dependent variables were all-cause death (either in-hospital or at 365 days). Covariates were added sequentially based on clinical relevance, previous research[11] and significant univariate associations (p<0.10), starting with age and sex and building a series of models to explore alternative risk-adjustment approaches. We generated time series of risk-adjusted CFRs by calculating standardised mortality ratios (SMRs) for each time period, dividing the number of observed deaths by the number of expected deaths (sum of probabilities from relevant model). Each model used pooled data across the 10-year period to allow comparisons over time. Risk-adjusted rates were calculated by multiplying the crude rate across 10 years by the SMR for each period (equivalent to indirect standardisation). The C-statistic and Akaike information criterion (AIC) values were calculated to assess model fit.

Fourth, we tested significance of national time trends by introducing 5-year period or fiscal year of discharge into models. For year-on-year trends, the year variable was treated as continuous, hence ORs reflect average change per 1 year increment across the decade (assuming a linear trend). Sensitivity analyses were undertaken using variants of model covariates (continuous rather than categorical variables for age and non-liver comorbidity score; recorded-stage of liver disease rather than binary variable).

Finally, we undertook exploratory analysis of regional variation focusing on the main outcome of in-hospital case fatality. CPRD region of residence was based on Strategic Health Authority boundaries. We used Funnel plots to evaluate variation in crude rates and added region to models to compare risk-adjusted CFR.

All analyses were undertaken in R (V.4.1.2, R Core Team 2021, R Foundation for Statistical Computing, Vienna, Austria).

### Patient and public involvement

Patients or the public were not involved in the design, conduct, reporting or dissemination plans of our research.

## RESULTS

### Characteristics of first emergency admissions

The algorithm detected 17 575 index emergency admissions with characteristics summarised in table 1

Table 1  Demographics, clinical characteristics, care processes and outcomes of patients admitted as an emergency for the first time with alcohol-related liver disease between 2008/2009 and 2017/2018

| Characteristic | Total (n=17 575) | Case definition | | P Value |
| | | Primary (n=9374) | Uplift (n=8201) | |
| --- | --- | --- | --- | --- |
| Age, mean (SD) | 53.4 (12) | 53.2 (12) | 53.6 (13) | * |
| Sex | | | | |
| Female | 5820 (33.1%) | 3335 (35.6%) | 2485 (30.3%) | ** |
| Male | 11 755 (66.9%) | 6039 (64.4%) | 5716 (69.7%) | |
| Deprivation quintile | | | | |
| 1—least deprived | 2435 (13.9%) | 1330 (14.2%) | 1105 (13.5%) | ns |
| 2 | 2735 (15.6%) | 1481 (15.8%) | 1254 (15.3%) | |
| 3 | 3028 (17.2%) | 1613 (17.2%) | 1415 (17.3%) | |
| 4 | 3837 (21.8%) | 2077 (22.2%) | 1760 (21.5%) | |
| 5—most deprived | 5540 (31.5%) | 2873 (30.6%) | 2667 (32.5%) | |
| Stage of liver disease | | | | |
| Fatty liver | 942 (5.4%) | 182 (1.9%) | 760 (9.3%) | ** |
| Hepatitis | 2369 (13.5%) | 1729 (18.4%) | 640 (7.8%) | |
| Fibrosis and sclerosis | 62 (0.4%) | 28 (0.3%) | 34 (0.4%) | |
| Cirrhosis | 5001 (28.5%) | 2759 (29.4%) | 2242 (27.3%) | |
| Hepatic failure | 2354 (13.4%) | 1982 (21.1%) | 372 (4.5%) | |
| Not specified | 6847 (39.0%) | 2694 (28.7%) | 4153 (50.6%) | |
| Ascites | 6589 (37.5%) | 4648 (49.6%) | 1941 (23.7%) | ** |
| Varices | 2704 (15.4%) | 1508 (16.1%) | 1196 (14.6%) | * |
| Comorbidity score (non-liver) | | | | |
| 0–1 | 15 506 (88.2%) | 8440 (90.0%) | 7066 (86.2%) | * |
| 2+ | 2069 (11.8%) | 934 (10.0%) | 1135 (13.8%) | |
| Five year period of discharge | | | | |
| 2008/2009–2012/2013 | 7566 (43.0%) | 4102 (43.8%) | 3464 (42.2%) | * |
| 2013/2014–2017/2018 | 10 009 (57.0%) | 5272 (56.2%) | 4737 (57.8%) | |
| GP contact in last year | 9936 (56.5%) | 5346 (57.0%) | 4590 (56.0%) | ns |
| With alcohol codes recorded† | 4702 (47.3%) | 2528 (47.3%) | 2174 (47.4%) | ns |
| With liver codes recorded† | 1465 (14.7%) | 865 (16.2%) | 600 (13.1%) | ** |
| Emergency admission in last year | 7265 (41.3%) | 3240 (34.6%) | 4025 (49.1%) | * |
| With alcohol codes recorded‡ | 5525 (76.0%) | 2393 (73.9%) | 3132 (77.8%) | ** |
| Length of stay, mean (SD) | 12 (15) | 14 (16) | 9 (13) | ** |
| Higher level care during admission | 1750 (10.0%) | 970 (10.3%) | 780 (9.5%) | ns |
| Died in-hospital | 2708 (15.4%) | 1779 (19.0%) | 929 (11.3%) | ** |
| Died within 365 days of admission | 5938 (33.8%) | 3642 (38.9%) | 2296 (28.0%) | ** |

Patients registered with an English CPRD practice. Data are shown for the cohort overall ('Total') and by case definition based on the Liverpool ARLD Algorithm ('Primary' or 'Uplift').
*p<0.05; **p<0.001 between primary and uplift groups.
†Percentages based on those with a GP contact within the last year.
‡Percentages based on those with an emergency admission within the last year.
ARLD, alcohol-related liver disease; CPRD, Clinical Practice Research Datalink; GP, General Practice; ns, not statistically significant.

(overall, and by case definition). Nine thousand three hundred and seventy-four (53%) were in the primary subgroup and 8201 (47%) in the uplift subgroup (non-primary coding patterns). The magnitude of 'uplift' is consistent with previous findings.[11 15]

The mean age was 53.4 years (range: 18–98), one-third (33.1%) were women, 32% were residents of most deprived quintile. Less than 5% had non-white ethnicity recorded (small numbers preclude more detailed reporting).

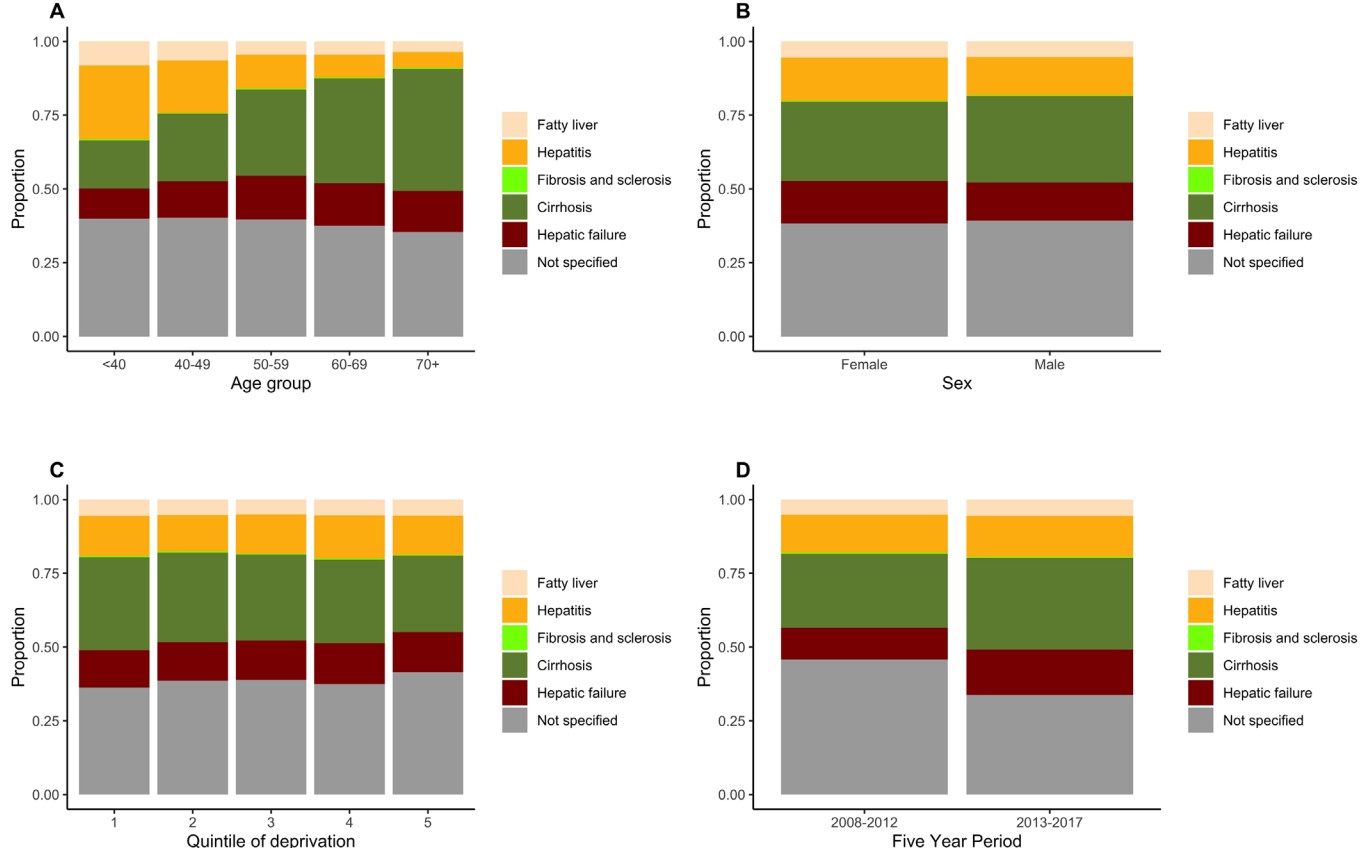

**Figure 1** Distribution of stage of alcohol-related liver disease at first emergency admission identified from the CPRD population of England. Stage is based on the ICD-10 codes recorded on discharge (n=17 575 cases). Stratified by: (A) age group, (B) sex, (C) deprivation quintile (1=least, 5=most deprived areas) and (D) five-year period of discharge. All p values <0.001 ($\chi^2$). CPRD, Clinical Practice Research Datalink; ICD, International Classification of Diseases-10.

## Prior primary care consultations

Nine thousand nine hundred and thirty-six patients had a record of ≥1 General Practitioner (GP) consultation in the year before admission (table 1). Of this subgroup, 4702 (47.3%) had a READ code relating to either alcohol consumption or any alcohol-specific condition recorded at that time, suggesting harmful drinking remained undetected or unrecorded in just over half of those encounters. Only 1465 (14.7% of 'recent' GP consulters) had a code relevant to liver disease recorded at the time, suggesting that ARLD usually remained unsuspected or undiagnosed during these consultations.

## Prior emergency admissions

Forty-one per cent of the cohort (7265 patients) had been admitted as an emergency in preceding year. Of these, 5525 (76%) had a non-liver-related, alcohol-specific condition recorded in their list of discharge diagnoses. Hence, one in four (24%) admitted cases had no record of an alcohol-related diagnosis in their preceding emergency admission. This was equally true of cases with cirrhosis or liver failure at index admission, suggesting serious alcohol-related harm may have gone unrecognised (table 1).

## Stage of liver disease

Stratification by recorded-stage is shown in figure 1 and online supplemental table S1. These data demonstrate the high proportion of advanced liver disease recorded at index hospitalisation (cirrhosis: 28%; hepatic failure: 13%). We found frequent use of K70.9 (alcoholic liver disease, unspecified) to encode first admissions, whereby disease stage was not specified in 39%. The presence of additional codes for ascites (31%) and varices (12%) in many such cases suggests a significant proportion had severe disease (online supplemental table S1).

Older groups had lower proportions with alcoholic hepatitis and a higher proportion with cirrhosis, as would be expected (figure 1A). The proportion with 'severe liver disease' codes (cirrhosis or hepatic failure) was similar among men and women despite statistical differences overall (figure 1B). Interestingly, the proportion of first admissions with severe disease showed limited variation across deprivation quintiles and was slightly lower among those from most deprived areas (figure 1C).

## Time trends in case-mix

We found small differences in sociodemographics between patients discharged in period 2 compared with

period 1 (online supplemental table S2). Mean age was 1 year older (54.1 vs 52.5 years, p<0.001) and there was a slight reduction in proportion from most deprived areas (30% vs 33%, p<0.001). There was an increased contribution of cirrhosis (31% vs 25%, p<0.001) and hepatic failure (15% vs 11%, p<0.05) but a reduction in unspecified stage (34% vs 46%, p<0.001) (figure 1D). There was also a slight increase in proportion with codes for varices (16% vs 15%, p=0.031). Levels of non-liver comorbidity were also higher (13% vs 10% with score ≥2, p<0.001). Collectively, these findings suggest patients at index admission were presenting with somewhat more advanced disease and greater burden of comorbidities.

Among patients with prior GP contacts (preceding year), we found no change in the proportion of consultations with alcohol-related READ codes (46.6% vs 48.1%, p=0.13) and a small reduction in liver disease codes in the final period (13.7% vs 16.0%, p<0.001). There was no change in recording of alcohol-specific codes in prior emergency admissions (75.6% vs 76.7%, p=0.3).

The characteristics of patients discharged within each fiscal year are summarised in online supplemental table S3.

### Care processes during first admission

Overall mean length of stay was 12 days (table 1), with higher duration for hepatic failure (16 days, p<0.001). Ten per cent received higher level care (21% of those coded as hepatic failure). Length of stay was 1 day shorter in the second half of the decade (11 days vs 12 days, p<0.001, online supplemental table S2), and a significantly greater proportion received higher level care (11% vs 8.6%, p<0.001).

### Deaths following first admission

Overall, 2708 deaths were recorded giving a crude CFR of 15.4% (table 1). Median time to in-hospital death was 9 days (mean: 14). One in five deaths occurred before day three of admission. Thirty-nine per cent of cases with a recorded-stage of hepatic failure died in hospital. Within 365 days of admission, the cumulative death toll rose to 5938 patients (34%).

Of the total in-hospital deaths, 2569 (95%) had descriptors listed in one or more fields that were consistent with an ARLD-specific cause. The most common descriptor was 'alcoholic liver disease, unspecified' (online supplemental appendix 2). Of these deaths, 1738 (66.8%) were in the primary group and 831 (32.2%) in the uplift group.

Among patients with GP consultations in preceding year (n=9936), those dying in hospital were less likely to have had coding of alcohol misuse or harms (39.3% vs 48.8%, p<0.01) or liver disease (10.1% vs 15.5%, p<0.05) than survivors. Patients with fatal admission were less likely to have had an emergency hospitalisation in the past year than survivors (35.9% vs 42.2%, p<0.01). However, among those previously hospitalised, alcohol-specific diagnoses were recorded less often in those subsequently dying than

survivors (68.7% vs 77.2%, p<0.01). Only 36.1% of in-hospital deaths received higher level care.

### Factors associated with death

Characteristics of patients who had died, or survived, at the two study end-points are shown in online supplemental table S4. In bivariate analyses we found significant associations with increasing age, female sex, primary (as opposed to uplift) case definition, increased non-liver comorbidity, advanced recorded-stage of liver disease and coding for ascites or varices. Stratified survival analyses confirmed these bivariate associations (online supplemental file 1). Multivariable models confirmed independent associations between the selected case-mix factors and odds of death (table 2). Interestingly, deprivation status (IMD quintile) was not independently associated with outcomes and not retained in models— suggesting severity of illness at presentation dominated prognosis. Given the high prevalence of unspecified stage we used a binary variable for recorded-stage (Liver failure or Not) for final risk-adjustment.

### Time trends in case fatality
#### Five-year periods

Comparison of unadjusted outcomes between period 1 and period 2 revealed lower crude in-hospital CFR (14.8% vs 16.2%, p=0.018). This did not quite reach significance at 365 days (33.2% vs 35.4%, p=0.063). Unadjusted rates of in-hospital death were numerically lower for each recorded-stage but not all comparisons were significant (hepatitis: 8.9% vs 9.9%, p=0.4; cirrhosis: 14.4% vs 17.7%, p=0.002; hepatic failure: 37.4% vs 41.2%, p=0.07; not specified: 9.7% vs 12.5%, p<0.001). Similarly, reductions in crude CFRs at 365 days did not all reach significance (hepatitis: 22.8% vs 24.5%, p=0.4; cirrhosis: 38.9% vs 41.9%, p=0.035; hepatic failure: 54.5% vs 57.7%, p=0.14; not specified: 26.0% vs 30.3%, p<0.001). Survival probabilities during the year after index admission were consistently higher for those discharged in period 2 period for the cohort as a whole and by recorded stage (figure 2). After adjusting for case-mix factors in relevant models, the adjusted odds of both in-hospital and 365 day death were significantly lower for those discharged in period 2 (table 2).

#### Year-to-year trends

Time series are shown in figure 3. Overall, crude in-hospital CFR declined from 18.3% in 2008/2009 to 16.1% in 2017/2018 (p=0.03) (figure 3A). Corresponding changes in stage-specific crude rates were 45.4% to 41.0% for hepatic failure (p=0.13), 21.1% to 13.0% for cirrhosis (p=0.0006) and 16.1% to 8.8% for alcoholic hepatitis (p=0.47). Case-mix-adjusted rates for in-hospital death showed significant reductions overall and in stage-specific analyses (figure 3B and table 2). In the final risk-adjusted model (table 2), each 1 year increment across the decade was associated (on average) with a 6% reduction in odds of in-hospital death (adjusted OR (aOR): 0.94 (95% CI:

**Table 2** Factors associated with death following first emergency admission for alcohol-related liver disease among people registered with CPRD practices in England, 2008/2009 to 2017/2018

| Covariate | Death during first admission | | | | Death within 365 days | | | |
|---|---|---|---|---|---|---|---|---|
| | OR | 95% CI | aOR | 95% CI | OR | 95% CI | aOR | 95% CI |
| Age group (ref: <40 years) | | | | | | | | |
| 40–49 | **1.42** | 1.18 to 1.70 | **1.31** | 1.09 to 1.58 | **1.33** | 1.18 to 1.50 | **1.2** | 1.06 to 1.37 |
| 50–59 | **2.36** | 1.99 to 2.81 | **2.02** | 1.70 to 2.43 | **2.01** | 1.79 to 2.26 | **1.67** | 1.48 to 1.88 |
| 60–69 | **3.39** | 2.85 to 4.04 | **2.86** | 2.39 to 3.43 | **2.97** | 2.63 to 3.35 | **2.36** | 2.08 to 2.68 |
| 70+ | **4.11** | 3.41 to 4.98 | **3.33** | 2.73 to 4.07 | **5.39** | 4.69 to 6.20 | **4.14** | 3.57 to 4.80 |
| Sex (ref: female) | | | | | | | | |
| Male | **0.86** | 0.79 to 0.93 | **0.79** | 0.72 to 0.87 | 1.00 | 0.94 to 1.07 | **0.93** | 0.86 to 0.99 |
| Case definition (ref: primary) | | | | | | | | |
| Uplift | **0.55** | 0.50 to 0.59 | **0.76** | 0.69 to 0.84 | **0.61** | 0.57 to 0.65 | **0.75** | 0.70 to 0.81 |
| Non-liver comorbidity (ref: 0–1) | | | | | | | | |
| 2+ | **2.38** | 2.14 to 2.64 | **1.97** | 1.75 to 2.21 | **2.92** | 2.66 to 3.20 | **1.97** | 1.75 to 2.21 |
| Hepatic failure (ref: any other stage) | | | | | | | | |
| Hepatic failure | **4.73** | 4.29 to 5.21 | **4.03** | 3.64 to 4.48 | **2.87** | 2.62 to 3.13 | **2.32** | 2.11 to 2.55 |
| Ascites (ref: no) | | | | | | | | |
| Yes | **2.03** | 1.87 to 2.21 | **1.48** | 1.35 to 1.62 | **2.31** | 2.17 to 2.47 | **1.86** | 1.74 to 2.00 |
| Varices (ref: no) | | | | | | | | |
| Yes | **1.35** | 1.21 to 1.50 | **1.2** | 1.07 to 1.34 | **1.41** | 1.30 to 1.53 | **1.21** | 1.11 to 1.33 |
| | Model 1, C-statistic: 0.733 | | | | Model 2, C-statistic: 0.710 | | | |
| Fiscal year (per year) | | | | | | | | |
| | **0.98** | 0.96 to 0.99 | **0.94** | 0.93 to 0.96 | **0.98** | 0.97 to 0.99 | **0.96** | 0.95 to 0.97 |
| | Fiscal year added to Model 1, C-statistic: 0.737 | | | | Fiscal year added to Model 2, C-statistic: 0.710 | | | |
| Five-year period (ref: 2008/2009 to 2012/2013) | | | | | | | | |
| 2013/2014 to 2017/2018 | **0.91** | 0.83 to 0.98 | **0.76** | 0.69 to 0.83 | **0.91** | 0.83 to 0.98 | **0.83** | 0.77 to 0.88 |
| | Five-year period added to Model 1, C-statistic: 0.736 | | | | Five-year period added to Model 2, C-statistic: 0.708 | | | |

The final case-mix adjustment models included seven patient-level baseline covariates. Additional time-related variables were then added to investigate time trends, using either fiscal year of discharge (continuous) or 5-year time period. Figure 3 illustrates the time trends in crude and case-mix-adjusted case fatality rates derived using these models.
Significant ORs are in bold text (p values <0.001 in all cases).
aOR, risk-adjusted OR; CPRD, Clinical Practice Research Datalink.

0.93 to 0.96), p<0.001). Stage-specific reductions in odds of dying during first admission were 3% per year for hepatic failure (p=0.004), 6% for cirrhosis (p<0.001) and 7% for alcoholic hepatitis (p<0.01).

Corresponding trends for 365-day death are shown (figure 3C,D). Overall crude rate reduced from 36.8% to 32.2% (p=0.02) and stage-specific crude rates showed downward trends—hepatic failure reducing from 59.2% to 53.9% (p=0.07), cirrhosis from 46.1% to 36.3% (p=0.02) and alcoholic hepatitis from 28.3% to 21.7% (p=0.15). After case-mix adjustment, each 1 year increment was associated with a 4% reduction in odds of dying within a year of admission overall (aOR: 0.96 (95% CI: 0.95 to 0.97), p<0.001). Stage-specific reductions in adjusted odds of death were also 4% per year for hepatic

failure (p=0.011), cirrhosis (p<0.001) and alcoholic hepatitis (p=0.027).

### Sensitivity analyses
Key findings were unchanged in terms of the significance or magnitude of trends when risk-adjustment models were fitted using variants of patient-level covariates (eg, continuous variables for age or comorbidity scores, individual categories for each stage of liver disease) (online supplemental figure S2).

### Regional variations in CFR of first admissions
Online supplemental table S5 shows distribution of cases and case-mix across the 10 regions of England. Crude CFR varied from 13.4% (London) to 20.1% (South East

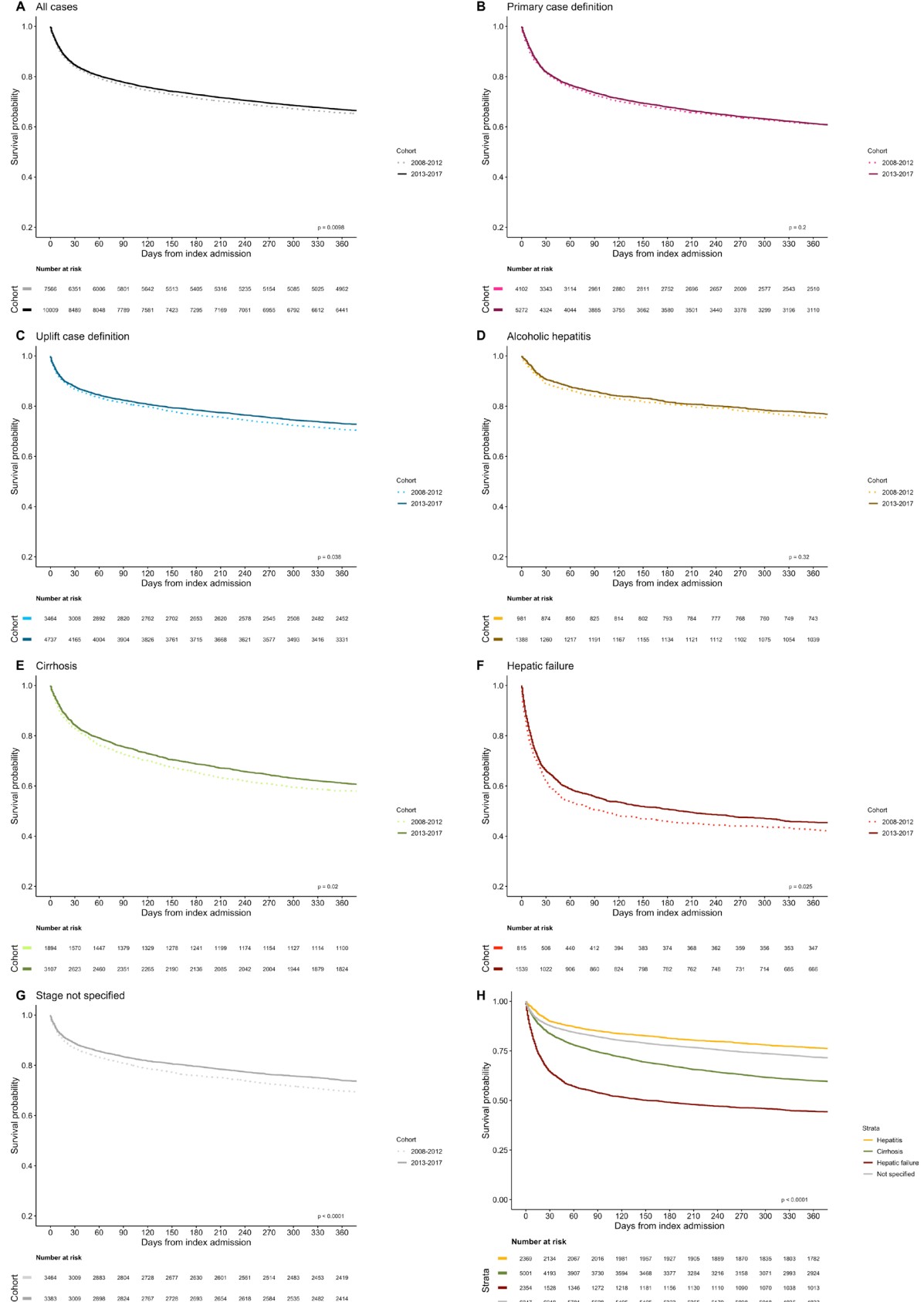

**Figure 2**  Kaplan-Meier survival probability following first emergency admission for alcohol-related liver disease. (A) to (G) are stratified into 5-year cohorts (2008/2009 to 2012/2013 vs 20013/2014 to 2017/2018). (A) Overall (n=17 575), (B) primary case definition (n=9374), (C) uplift case definition (n=8201), (D) alcoholic hepatitis (n=2369), (E) cirrhosis (n=5001), (F) hepatic failure (n=2354), (G) unspecified stage of liver disease (n=6847). (H) shows all cases stratified by recorded-stage of liver disease. P values represent significance level of log rank test.

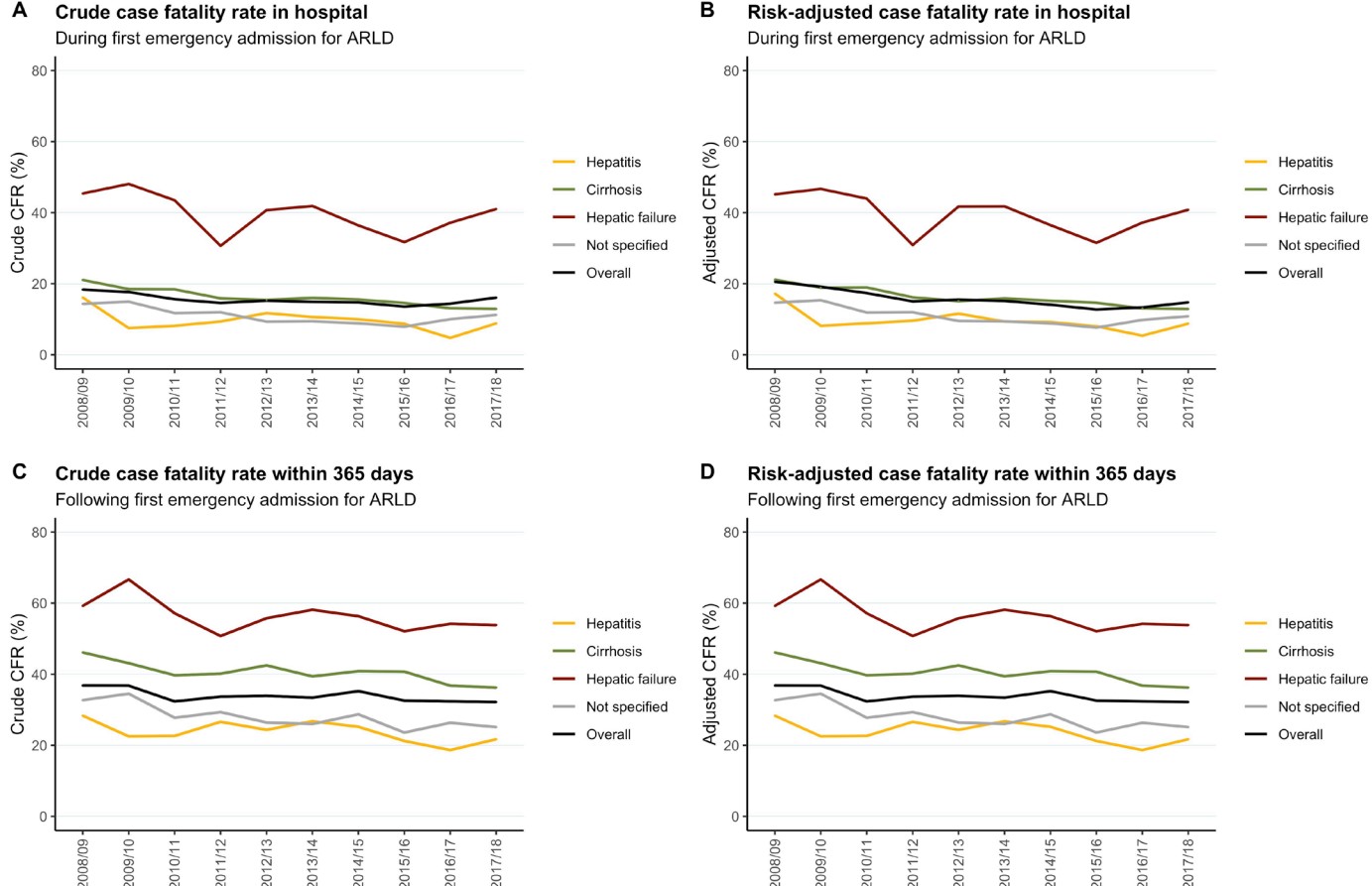

**Figure 3** Time trends in case fatality rates (CFRs) for first emergency admissions for alcohol-related liver disease, overall and by stage of liver disease. (A) Crude CFR in-hospital, (B) risk-adjusted CFR in-hospital, (C) crude CFR within 365 days, (D) risk-adjusted CFR within 365 days. Risk-adjustment for overall rates was by age, sex, case definition (primary or uplift), comorbidity score (non-liver), stage of liver disease (liver failure or not) and coding for ascites and varices. Binary logistic regression models confirmed significant downward linear trends over the 10 years overall, before and after risk-adjustment (see table 2). For stage-specific rates, risk-adjustment was by age, sex, case definition and comorbidity score, with additional relevant models confirming significance of trends. ARLD, alcohol-related liver disease.

Coast). Funnel charts identified two regions as potential 'outliers' (ie, higher than the average) on the basis of unadjusted rates—the West Midlands and South East Coast (online supplemental figure S3)—whereas London had a lower-than-average crude rate. When region was added to relevant models (online supplemental table S6), these two parts of the country had a significantly increased odds of in-hospital death relative to London, with the West Midlands having an OR of 1.27 (95% CI: 1.09 to 1.49) and South East Coast an OR of 1.36 (95% CI: 1.12 to 1.66). There was no significant difference between the other regions. Overall national time trends were not altered by including region in the 5-year and year-to-year models.

## DISCUSSION

Our study represents the first detailed description of characteristics and outcomes of index emergency admissions with ARLD in the English population. The predominance of men and disproportionate contribution of people living in more deprived areas is consistent with known epidemiological patterns.[1] These admissions encompass diverse manifestations of liver disease and alcohol misuse, ranging from early-stage liver disease to rapidly fatal liver failure. However, at least half had advanced disease.

Our data suggest ARLD is frequently diagnosed for the first time during unplanned hospitalisation. Only 14.7% of cases with GP contacts in the year before index emergency admission for ARLD had liver disease recorded during those encounters, suggesting underlying liver problems were not suspected or diagnosed at that time. A previous study examined the earliest instance of coding for cirrhosis (regardless of aetiology) within linked CPRD-HES records (1998–2009), reporting this occurred first within an emergency record in half the cases.[21] We examined recent GP consultations for codes relating to alcohol excess, finding them absent in half. These data imply missed opportunities to screen for, or detect, alcohol-related harms at a point when patients would be expected to have had well-established harmful drinking.

Similar issues were apparent in secondary care with alcohol-related codes absent from almost a quarter of

unplanned hospitalisations in preceding year. The corresponding figure was 31.3% for patients who subsequently died in-hospital. It seems unlikely such patients lacked evidence of alcohol-specific conditions during those prior admissions. Across the decade, we found no improvement in rates of recognition/recording of alcohol-related problems in primary or secondary care contacts during the year before index hospitalisation. Deficiencies in early detection have been recognised internationally.[22]

We found improvements in short-term and medium-term prognosis of index emergency admissions over time but the magnitude was small. In 2017/2018, one in seven were still dying during their first hospitalisation, rising to one-third within a year. While survival was better in the post-NCEPOD era, yearly time series suggested no step-change or accelerated reduction in case fatality during this sustained period of national initiatives.[2–5 7] Directly comparable statistics for other countries are lacking.

Only a third of in-hospital deaths (36%) had received higher level care. This is comparable to 39% in the NCEPOD review where 31.2% of those deemed eligible (by reviewers) for higher level care did not receive it.[2] This prompted recommendations for improved decision-making and access. Universal prognostic pessimism about organ support in severe liver disease has been challenged.[23] We cannot judge suitability of cases for escalation in our study but we did find a small increase in the proportion receiving such care over a 10-year period.

Exploratory analyses suggest two regions had higher crude CFRs overall and differences persisted after risk-adjustment. However, most regions were comparable on these measures. Caution is needed when interpreting local-level outcome metrics derived from associations in national data (constant-risk fallacy). Geographical variation in population-based mortality rates (as opposed to CFRs) for liver disease are well-recognised but further exploration of possible inequalities in outcome of first admissions per se is merited.[24]

We confirm limitations of simple case definitions for defining workload and outcome.[11 15 16 21] Recent studies of international burden of 'decompensated cirrhosis' continue to rely on primary discharge diagnosis.[25] Our analysis of deaths shows that many of the sickest patients had a non-primary coding pattern. Our findings on 'recorded-stage' question the merits of defining 'severe' liver disease based solely on primary codes (cirrhosis or liver failure).[26] Unspecified stage accounted for 28% of admissions and one in four deaths. If administrative data are to be used to quantify burden, evaluate interventions and benchmark performance then case identification and risk-adjustment requires care.[16] The LAA-based approach,[11 15] combined with case-mix classification reported here, could support development of improved metrics as recommended by the *Lancet* commission and in a recent NCEPOD update.[13 27] Existing statistics require caveats for interpretation.[24]

Our study has several limitations. First, the available data allowed us to define 'first' emergency admission based on an admission-free interval of 10 years. We cannot rule out inclusion of patients admitted as an emegency over a decade earlier who then avoided any subsequent emergency readmission or death. Given the prognosis and typical trajectory of ARLD, we believe such cases are rare.

Second, HES data lack true disease severity measures (eg, clinical scores or laboratory parameters), limiting risk-adjustment. However, we applied both recorded-stage and condition-specific markers (ascites and varices) in addition to the usual generic covariates. We omitted hepatic encephalopathy—an important complication of severe disease[14]—based on experience of LAA development.[11] We found relevant codes (G92X toxic encephalopathy; G934 encephalopathy unspecified) were seldom recorded. Codes for altered consciousness or confusion were common but could indicate alcohol intoxication or withdrawal. Only 441 cases (2.5%) in the present study had encephalopathy coded, of which just 164 (<1%) lacked additional coding for ascites or varices.

Over the decade, first admissions appeared to have somewhat more severe disease and higher burden of comorbidities. The possibility of changing coding practices over time cannot be excluded in any long-term study of administrative data. However, our time trends for CFR were significant both for crude data and in sensitivity analyses omitting relevant covariates which suggests prognostic improvement is genuine rather than artefactual.

Third, while CPRD practices serve a representative population of English residents,[17] our results are not generalisable to every locality. A key aim was to examine events in primary care during the year before index admission but linked data are not available for the entire country. Further work using complete national HES is needed to quantify national trends and explore regional and provider variation.

Finally, although LAA performance in detecting true cases of ARLD against hospital records has been reported previously,[11 15] we could not conduct formal validity testing as inpatient data was limited to HES. However, linkage to death certification suggested the majority of in-hospital deaths in the uplift subgroup were related to alcoholic liver disease. The LAA is based on coding patterns observed in English HES data. It is unknown whether such variations exist in other countries.

In conclusion, we found evidence of gradual improvement in prognosis for first emergency admissions in England over a 10-year period. Nevertheless, at least one in seven patients were still dying during index hospitalisation and a third within a year. Many patients had contacts with health services in the year prior to admission but without apparent recognition of alcohol-related harms, adding to concerns about missed opportunities for screening and intervention.[20] Geographical variations require further investigation. Our observations support much of the pessimism expressed in the *Lancet Commission*'s final report[10] This highlights the need for concerted action in public health interventions and primary and secondary care to improve prevention, early

detection and inpatient treatment of alcohol-related harms.

**Contributors** KB is the guarantor of the article and responsible for the overall content. KB wrote the manuscript and all authors reviewed and approved the final submission. KB, KMF, BS and PS designed the research with input from TM and SH. KB, BS and TM curated and analysed the data with input from PS and KMF.

**Funding** This work was funded by the UK Department of Health (Connected Health Cities programme) and delivered by the Northern Health Science Alliance. The funding sponsors had no role in the design and conduct of the study; in the collection, management, analysis and interpretation of the data; or in the preparation, review or approval of the manuscript.

**Competing interests** None declared.

**Patient and public involvement** Patients and/or the public were not involved in the design, or conduct, or reporting, or dissemination plans of this research.

**Patient consent for publication** Not applicable.

**Ethics approval** This study involves human participants but the CPRD has obtained ethical approval from the UK's National Research Ethics Service (NRES) Committee for observational research using anonymised data. Our study was approved by the Independent Scientific Advisory Committee (ISAC) for Medicines and Healthcare Products Regulatory Agency (MHRA) database research (protocol number 19_133). Patient consent for publication is not applicable for approved studies using the CPRD dataset.

**Provenance and peer review** Not commissioned; externally peer reviewed.

**Data availability statement** No data are available. This study is based in part on data from the Clinical Practice Research Datalink obtained under licence from the UK Medicines and Healthcare products Regulatory Agency. The data are provided by patients and collected by the NHS as part of their care and support. The interpretation and conclusions contained in this study are those of the author/s alone. Copyright ©2023, re-used with the permission of The Health & Social Care Information Centre. All rights reserved. CPRD data are not publicly or freely available but can be obtained under special licence (https://cprd.com/data-access)

**ORCID iD**
Keith Bodger http://orcid.org/0000-0002-1825-3239

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
