## [Reviewer comments · BMJ Open]

ARTICLE DETAILS

TITLE (PROVISIONAL)	Outcomes of first emergency admissions for alcohol-related liver disease in England over a ten year period: Retrospective observational cohort study using linked electronic databases
AUTHORS	Bodger, Keith; Mair, Thomas; Schofield, Peit�; Silberberg, Benjamin; Hood, Steve; Fleming, Kate M

VERSION 1 – REVIEW

REVIEWER	Bernal, William King's College Hospital
REVIEW RETURNED	23-Jul-2023

GENERAL COMMENTS	I reviewed this report with Dr Jessica King (LSHTM). This is a generally well-written and timely report addressing an important subject. Its findings have clear clinical implications and reinforce the key importance of the earlier detection of alcohol-related harm in patient contacts in both primary and secondary care – whether or not these relate to liver disease. There is however opportunity for improvement, by example through more detail around definitions and clarification of the reported results, and through altered emphasis on specific and significant limitations. Amongst these is that inherent to the use of CPRD: as vividly demonstrated in the Atlas of Variation of Liver Disease (uncited in this report) there is major national geographic variation in liver disease incidence and outcome and these important regional effects are unexplored. A further and similarly important limitation is the assignment of liver disease severity based upon primary codes – accurate assessment of liver disease severity is a challenge for practicing hepatologists, and extrapolation from HES codes even more uncertain – particularly since more than a quarter of cases had an unspecified stage of ARLD. The authors mention both these limitations in the discussion, but I felt these warranted greater emphasis. I also felt that a greater emphasis should be placed upon a key message of the report – ‘making every contact count’ through active screening for problem alcohol use. There unacceptable proportion of cases presenting for the first time with advanced stage disease even when there have been prior contacts in primary and / or secondary care with other medical issues. As part of this discussion, it would be helpful for the more information on the natural history of ARLD. How long would a patient need to have been using alcohol harmfully to develop ARLD at the various stages you describe? How long are the earlier stages of ARLD and what symptoms might be being missed by GP/previous hospital visits? Other Comments: P4 ll18-19. Please can you clarify whether a previous
---

	planned/elective admission for ARLD would exclude a patient from your cohort or not. If not, there are presumably patients in your cohort who do have a previous diagnosis of ALRD, just not as an emergency. What is your justification for their inclusion in the cohort? P4 l126-38. It would be helpful to include the full LAA definitions in the appendix as well as referring to the APT papers. P4 l59 please specify the codes (perhaps in appendix) used to define varices, ascites and comorbidities of interest. P5 l13 why only one year? P5 l25 Why was this limited to unplanned care? What is the justification for examining only a year's previous admissions? P6 l33 This is the only place you mention linkage to intensive care data. Please can you expand on this a little more - is this the Critical Care Minimum Dataset? P6 l4-8 Please specify (perhaps in appendix) a) which codes were used and b) whether only underlying cause of death or any cause of death was used. P7 l41-44 'these figures support the assertion..' this is interpretive and belongs in discussion, not results. P7 l50-52 "This leaves almost one in four with no preceding record of an alcohol-related diagnosis (despite subsequently being readmitted as an emergency with ARLD)" I have a few concerns with this statement:  -The group with no preceding record of alcohol-related emergency admission in the last year are the 59% who have no emergency admission in the preceding year at all, plus the 24%\times41%\approx9% who had an emergency admission, so ~68% of the total. -This isn't the same as saying they have no preceding record of an alcohol-related diagnosis: what about primary care and non-emergency admissions? What about diagnoses more than a year before the index admission? -In any case this kind of interpretive statement is again better in the discussion P7 l58 onwards. Stage of liver disease – see comments above. It would be supportive for the analyses that follow to move the K-M plot in supplementary figure 1 D to the main body. P8 l14-21 This paragraph simply describes patterns in the figures without adding important information such as proportions and the results of statistical tests, which can only be found in the appendix table. Your statements "the proportion of first admissions with severe disease showed limited variation across deprivation quintiles" and "the proportion with "severe liver disease" (codes for cirrhosis or hepatic failure) was similar among men and women" seem to contradict the statistical testing reported in S1, which found a difference in the proportions significant at the $p=0.001$ level for both of these characteristics. Perhaps statistically significant but not large/important differences? P8 l23-34. Perhaps with these small differences it would be useful to give age and percentages to one rather than zero decimal places in this paragraph and arguably could do so throughout paper. P8 l34 P8 l34-40 'Collectively these findings...' Interpretive and better in the discussion, not results. If changing coding practices over time is a genuine concern, would like to see more in-depth discussion of this and how it impacts on interpretation in the discussion section. P8 l42 'Concerningly.' emotive and should be replaced. P8 l45. $P=0.13$ is really rather weak evidence for any change in the proportion of GP contacts with a record of alcohol, not sure it's valid
--	---

	to report it as a reduction. Separately, it would be useful to report the proportion of all patients who a GP contact with a record of alcohol/liver disease codes, i.e. with the denominator being all patients not just those with a GP contact in the last year. Otherwise, a decrease in the proportion with liver disease codes could for example be driven by an increase in proportion with a GP contact in last year, but no decrease in the overall recognition of liver disease P9 I57 How did you deal with patients transferred between hospitals when calculating length of stay? P9 I17 when you refer to deaths within 48 hours of admission, does this mean within 1 day of admission or 2 days of admission? Note that a death within 2 days of admission does not necessarily mean within 48 hours, since neither time of admission nor death is available in these data, e.g. patient admitted 3am Monday and died 3pm Wednesday. Use of 'just 48 hours' emotive. P9 I45-47 Or died before transfer to ICU? Suggest move this to discussion section anyway. P10 I5. I can't find a Table 3 in the main ms. P10 I14-16 Again think an additional decimal place would be useful on the CFRs P11 I5-10. Please put results of sensitivity analyses in appendix Discussion – see comments above.
--	--

REVIEWER	Brown, Cristal The University of Texas at Austin, Internal Medicine
REVIEW RETURNED	27-Jul-2023

GENERAL COMMENTS	Peer Review - Outcomes of first emergency admissions for alcohol-related liver disease in England over a ten year period: Retrospective observational cohort study using linked electronic databases  1. First reading – big picture  a. Significance of the research question  i. Research question is significant. Alcohol-associated liver disease is prevalent, underdiagnosed and associated with high morbidity and mortality. b. Originality of the research  i. This study uses nationwide, linked data and applies an innovative algorithm to more accurately identify patients with ARLD for characterization and reporting of mortality trends over time c. Writing style and structure  i. Writing style is appropriate such that readers unfamiliar with the topic can follow the methodology, results and conclusions ii. Ethical considerations are appropriately addressed. d. Summarize the paper in a few sentences or a paragraph  i. This study uses a novel algorithm, previously tested and manually reviewed with consistent accuracy, to identify patients with ARLD presenting for an initial emergency admission. Using linked data, the authors characterize the population and provide case fatality with description of trends in mortality over 10 years spanning 5 years prior to and after the NCEPOD report which prompted attention to improving care for patients with ARLD. This study concludes that there are continued missed opportunities in primary and secondary care to identify
--

these patients and ongoing high mortality rates in the first year following initial emergency admission.

2. Second reading

Look for possible improvements to clarify the presentation
Provide specific suggestions

a. **Abstract**

- i. Adequately summarizes the findings of the study and provides concise and pertinent information.

b. **Introduction**

- i. The authors do a good job of establishing the importance of the research topic. There has been data driven concern for the underdiagnosis and inadequate care for patients with ARLD. Strong citations of the NCEPOD and Lancet reports noting unacceptably high levels of mortality in this hospitalized patient population
- ii. These authors acknowledge the gap in the literature regarding characterization of this population on a nationwide level
- iii. They utilize an algorithm that they have previously tested and manually confirmed accuracy to improve the ability to detect patients with ARLD and link to available hospital administrative and death data.
- iv. Study aims are clearly defined.

c. **Methods**

- i. The research design is appropriate to address the study aims
- ii. Although there is sufficient description of the data sources, case definitions, and outcomes, I would recommend clarification of the following items:
 - 1. Under "Population of interest", provide an explanation for defining index admission as having no such admission within the preceding 10 years as opposed to no prior admissions at all.
 - 2. Under "Patient-level covariates derived from the index admission", why was hepatic encephalopathy not included as a co-variate for advanced liver disease? Ascites, variceal hemorrhage and HE are the 3 major complications associated with decompensated liver disease and the omission of this symptom should be addressed.
 - 3. Under "Patient-level covariates derived from the index admission", it would also be helpful to explain the decision to separate CCI categories of 0-1 and ≥ 2 . Is this standard or provide an explanation for this choice
- iii. There are no concerns with the statistical methods utilized and their methodological description.

d. **Results**

- i. The results are generally reported appropriately based on the stated objectives and methods.
- ii. I would consider the following:
 - 1. I have concerns regarding the use of stage of liver disease as a co-variate and analysis of its association with mortality. Almost 40% of the

	patients have unspecified stage of liver disease. This means that the true percentage of patients presenting with early and advanced liver disease cannot be known. The authors do acknowledge the high rate of unspecified stage and include the supplemental table 1 to suggest a reasonable portion have advanced disease based on the presence of ascites and varices but I would caution using this variable for reporting association in the time trends in case fatality. The multivariate model uses only advanced stages and this is appropriate given the coding issue.  2. In “Time trends in case-mix”, the last sentence of the first paragraph regarding the improvement in coding over time should be included in the discussion and not reported in results. 3. There are multiple instances when a non-significant p-value is reported as having a significant change or trend toward significance. These should be corrected.  a. Concerningly, among patients with prior GP contact in the preceding year we found reductions in the proportion of consultations with a record of alcohol (46.6% vs. 48.1%, p=0.13) b. There was also a small reduction in recording of alcohol-specific codes in prior emergency admissions (75.6% vs. 76.7%, p=0.3). c. This did not quite reach significance for the 365 day end-point (33% vs 35%, p=0.063) d. Overall crude rate reduced from 36.8% to 32.2% (p=0.02) and stage-specific crude rates showed downward trends – hepatic failure reducing from 59.2% to 53.9% (p=0.07), cirrhosis from 46.1% to 36.3% (p=0.02) and alcoholic hepatitis from 28.3% to 21.7% (p=0.15). 4. Under “Deaths following first admission”, the following sentence belongs in discussion and not results. “These data provide further indirect evidence to support the validity of the LAA for cohort discovery and highlight the risk of missing a substantial mortality burden using HES-based statistics derived using the traditional primary definition.” 5. If stage-specific outcomes for time trends are retained, I recommend separating those stages with significant p values from those with insignificant p values 6. Need to add p value at the end of this
--	--

	statement. “Stage-specific reductions in odds of dying during first admission were 3% per year for hepatic failure (p=0.004), 6% for cirrhosis (p<0.001) and 7% for alcoholic hepatitis”. e. Discussion  i. I would consider removing the first paragraph of the discussion. It is a restatement of the background which has been addressed in the introduction. ii. Also consider changing the order of the paragraphs in the discussion. Readers would like a summary of the important results first – related to the primary objectives of the discussion, followed by the importance of these results. It takes 3 to 4 paragraphs before you address the major findings. f. Conclusion  i. I agree with the conclusions and limitations stated by the authors. The study successfully achieves its objectives in finding some improvement in prognosis for patients with ARLD but with ongoing concerns for late diagnosis and need for improved hospital care. g. Figures and Tables  i. The figures and tables in the manuscript and supplemental materials are pertinent to the findings and are consistent with the reported results. They are appropriately described and consistently reported throughout. h. References  i. Relevant and up to date references provided
--	---

VERSION 1 – AUTHOR RESPONSE

Reviewer: 1

Dr. William Bernal, King's College Hospital

Comments to the Author:

“I reviewed this report with Dr Jessica King (LSHTM). This is a generally well-written and timely report addressing an important subject. Its findings have clear clinical implications and reinforce the key importance of the earlier detection of alcohol-related harm in patient contacts in both primary and secondary care – whether or not these relate to liver disease. There is however opportunity for improvement, by example through more detail around definitions and clarification of the reported results, and through altered emphasis on specific and significant limitations.”

Reply: We thank Professor Bernal for his review and for inviting further expert critique from Dr King. The Methods section has been expanded (and new supplementary materials added) to clarify each of the points listed under ‘Other Comments’ and we have addressed perceived limitations.

“Amongst these is that inherent to the use of CPRD: as vividly demonstrated in the Atlas of Variation of Liver Disease (uncited in this report) there is major national geographic variation in liver disease incidence and outcome and these important regional effects are unexplored.” HES-linked CPRD was the only dataset suited to the aims of the present study since we wished to examine primary care contacts during the year before index emergency hospital

admission to explore potentially “missed opportunities” in the care pathway. HES-linked CPRD

is the only source of nationally-representative linked data between primary care electronic records and HES.

Our work focused on reporting at “national” aggregate level for the CPRD population and has important implications for interpreting routine national statistics for advanced liver disease in the UK. We believe it is essential that methods for cohort discovery and characterisation of ARLD are improved and that “getting it right” at national-level is a prerequisite before drilling down to local levels. National trends are of most relevance to an international readership and we note Reviewer 2 did not raise this particular point.

We regard the full national HES dataset as better-suited to exploring local or institutional-level variation but it cannot be linked to primary care events. The “Atlas of Variation of Liver Disease” uses the full national HES dataset. It contains various metrics related to “cirrhosis” of any aetiology (not ARLD per se) and uses standard approaches to case definition. Based on our past work and data presented in the current paper, that data source requires careful interpretation. Such statistics do not take full account of the contribution of non-primary coding

patterns or unspecified stages of liver disease to true burden and mortality from “cirrhosis”. Nevertheless, we agree that there’s abundant evidence of geographical variation in the incidence of alcohol-related harms based on population-level statistics and of a strong association between prevalence with socioeconomic factors. Our present work is more focused on care processes and care outcomes (i.e. what happens to patients in the year leading up to admission and what is the outcome of acute hospitalization?). We show that our primary metric of interest (case-mix adjusted CFRs for first admissions) is not associated with area-based deprivation status per se but rather with patient-level markers of “severity”. This justifies our inference that observed national trends in this measure may reflect improvements in hospital care at this key milestone in the care pathway, albeit modest and without a major national shift in underlying trends post-NCEPOD etc.

2

However, to further address this comment we have added some focused exploratory regional analyses. These examine in-hospital case-fatality rate. The methods, results and discussion sections have been revised to include these new data. Word limits mean that most of this data has to appear in Supplementary Materials.

In our revised discussion, we further emphasize the limitations of CPRD and the need for further research. The present report already provides a large amount of novel national-level data. A more extensive exploration of regional variations is beyond the scope of a single paper. Further research is ongoing and we hope our work stimulates others to seek to improve routine statistics, such as the ‘Atlas of Variation’, by adopting algorithm-based methods.

“A further and similarly important limitation is the assignment of liver disease severity based upon primary codes – accurate assessment of liver disease severity is a challenge for practicing hepatologists, and extrapolation from HES codes even more uncertain – particularly since more than a quarter of cases had an unspecified stage of ARLD. The authors mention both these limitations in the discussion, but I felt these warranted greater emphasis.”

Firstly, we would emphasize that the assignment of severity was not determined from “primary codes”. This statement implies that staging was based exclusively on whichever ICD-Code had been recorded as the “primary” diagnosis (i.e. the code appearing in the first diagnosis field of HES). This was not the case. Our algorithmic method screened all coding positions to find the relevant liver-related code and takes the “recorded-stage” from this. The method for assigning recorded-stage takes advantage of the LAA, whereby the liver-related code can appear in a lower diagnostic position (provided the primary diagnosis is relevant to an emergency admission for ARLD). Hence, a patient with a primary diagnosis of “Jaundice” followed by a secondary diagnosis of “K703 Alcoholic cirrhosis of the liver” would be flagged as

an eligible case and “staged” as cirrhotic. Additionally, ‘severity’ of liver disease was reflected in risk-adjustment by screening all diagnostic positions for stage-specific ‘flags’ (Ascites and Varices). This approach moves beyond existing methods (in common use) for creating official national statistics whereby only the primary diagnosis is considered and half of true admissions may be ignored.

Secondly, we have revised the manuscript to emphasize the term “recorded-stage” which should help clarify what is a key strength, not a limitation, of the research. A key motivator of our past and present work has been to drive improvements in the way routine data (specifically,

HES) are interrogated and interpreted. The original algorithm (LAA) was generated to align with clinical realities and inevitable variations in hospital coding of emergency admissions for ARLD and improve cohort discovery. The present work builds on this research by examining the implications of relying on recorded-stage of liver disease for generating statistics.

We report the frequent recording of ‘unspecified’ (K70.9) code rather than a stage-specific code in HES in “first” admissions. This is entirely predictable from a clinical perspective (especially at “first” admission) but has significant implications for interpreting the literature and current public health metrics. Typically, cases coded with K70.9 are missing from cohort definitions of “advanced liver disease” or “cirrhosis” based on administrative data, such as the

the Atlas of Variation or published research. Within this often-excluded group, we show a significant prevalence of markers of advanced liver disease (ascites and varices). We demonstrate their contribution to overall case volume and mortality burden and present data on their survival relative to sub-populations with a stage-specific discharge code. These data are an essential component of the present paper, highlighting the limitations of making simplistic assumptions about disease stage and the potential merits of algorithmic approaches

for cohort discovery and phenotyping. For the final risk-adjustment models of case fatality the reviewers will note that we adopt a simplified binary variable for recorded-stage (Liver Failure vs All Other Codes) to mitigate this phenomenon and we use additional “flags” of liver disease severity derived from other codes (Ascites and Varices).

3

“I also felt that a greater emphasis should be placed upon a key message of the report – ‘making every contact count’ though active screening for problem alcohol use. There unacceptable proportion of cases presenting for the first time with advanced stage disease even when there have been prior contacts in primary and / or secondary care with other medical issues. As part of this discussion, it would be helpful for the more information on the natural history of ARLD. How long would a patient need to have been using alcohol harmfully to develop ARLD at the various stages you describe? How long are the earlier stages of ARLD and what symptoms might be being missed by GP/previous hospital visits?”

Reply: There are a number of key messages, including the positive findings of improving prognosis of admissions over time. However, we have revised the manuscript accordingly. Our conclusions in the abstract and final part of discussion place strong emphasis on earlier recognition and intervention, citing relevant literature for readers.

We have expanded the methods section to emphasize the rationale for focusing on events within a year of emergency admission for ARLD.

“Other Comments:

P4 I18-19. Please can you clarify whether a previous planned/elective admission for ARLD would exclude a patient from your cohort or not. If not, there are presumably patients in your cohort who do have a previous diagnosis of ARLD, just not as an emergency. What is your justification for their inclusion in the cohort?”

Reply: Our study focuses on first emergency admission for ARLD (i.e. non-elective). We were interested in studying the case mix and outcomes of patients hospitalized acutely for a

first time with this condition. We have proposed previously that a person's index unplanned hospitalization for ARLD represents a critical point in the care pathway and a pragmatic focus for developing better metrics of initial acute care derived from hospital administrative data. In the paper's title, and throughout the manuscript, we emphasize that our primary focus was on emergency admission. This key milestone in the patient journey may often represent the point of first-ever contact with liver services and/or the point of first-ever formal diagnosis of ARLD. However, our report focuses on the processes and outcomes of the index emergency admission for ARLD, as opposed to seeking to identify a subcohort of first emergency admissions which were a verifiable point of "first-ever" diagnosis. Identifying index emergency

admissions does not specifically require screening of elective admissions.

However, this comment is relevant to one of the sentences in our original discussion (paragraph 3), where we had stated that ARLD is 'frequently diagnosed for the first time during unplanned admission'. We have revised and better-qualified this statement.

There are only a limited number of reasons why a patient with established ARLD might be admitted "electively" to hospital for planned treatment of liver disease per se – e.g. either for daycase endoscopy (for management of varices) or daycase paracentesis (for treatment-resistant ascites). In our experience, such elective admissions are almost invariably downstream of one or more previous emergency presentations. Nevertheless, we accept that a patient's first emergency admission for ARLD might have been preceded by a non-elective admission in some cases. This does not alter the main focus of our paper (i.e. characteristics and outcomes of index emergency admissions).

We would expect prior discharge diagnoses of ARLD to be known and reflected in the primary care record. We report that only 14.7% of consulters in primary care during the year prior to admission had READ codes for liver disease (i.e. a prior elective diagnosis). We believe this would include patients with well-known liver disease who had received prior elective hospital

4

care (and yet escaped any prior emergency hospitalisation). However, we acknowledge the perceived limitation in our revision.

"P4 I126-38. It would be helpful to include the full LAA definitions in the appendix as well as referring to the APT papers."

Reply: The LAA code lists are published and freely-available "open access". The relevant link to download the code lists has been added to the manuscript. To encourage appropriate citation of our original work, we prefer to sign-post that existing link rather than replicating publication in another supplementary file.

<https://onlinelibrary.wiley.com/action/downloadSupplement?doi=10.1111%2Fapt.15781&file=apt15781-sup-0001-Supinfo.docx>

P4 I59 please specify the codes (perhaps in appendix) used to define varices, ascites and comorbidities of interest.

Reply: For the liver-specific complications the ICD-10 codes are listed in the LAA and are easily accessed using the original link (as above)

<https://onlinelibrary.wiley.com/action/downloadSupplement?doi=10.1111%2Fapt.15781&file=apt15781-sup-0001-Supinfo.docx>

For the Charlson Comorbidity Index (CCI), the codes are available from the source publication and we have added the citation to give appropriate attribution. We have updated the methods and references. To focus on non-liver comorbidity, we simply excluded "mild" liver disease and "moderate or severe" liver disease categories.

• Charlson ME, Pompei P, Ales KL, MacKenzie CR. A new method of classifying prognostic comorbidity in longitudinal studies: development and validation. *J Chronic Dis.* 1987;40(5):373-83. doi: 10.1016/0021-9681(87)90171-8. PMID: 3558716.

These extensively utilised code lists are given below (for reviewer's convenience):

Myocardial infarction: I21.x, I22.x, I25.2

Congestive heart failure: I09.9, I11.0, I13.0, I13.2, I25.5, I42.0, I42.5 - I42.9, I43.x, I50.x, P29.0
Peripheral vascular disease: I70.x, I71.x, I73.1, I73.8, I73.9, I77.1, I79.0, I79.2, K55.1, K55.8, K55.9, Z95.8, Z95.9
Cerebrovascular disease: G45.x, G46.x, H34.0, I60.x - I69.x
Dementia: F00.x - F03.x, F05.1, G30.x, G31.1
Chronic pulmonary disease: I27.8, I27.9, J40.x - J47.x, J60.x - J67.x, J68.4, J70.1, J70.3
Rheumatic disease: M05.x, M06.x, M31.5, M32.x - M34.x, M35.1, M35.3, M36.0
Peptic ulcer disease: K25.x - K28.x
Mild liver disease: B18.x, K70.0 - K70.3, K70.9, K71.3 - K71.5, K71.7, K73.x, K74.x, K76.0, K76.2 - K76.4, K76.8, K76.9, Z94.4

5

Diabetes without chronic complication: E10.0, E10.1, E10.6, E10.8, E10.9, E11.0, E11.1, E11.6, E11.8, E11.9, E12.0, E12.1, E12.6, E12.8, E12.9, E13.0, E13.1, E13.6, E13.8, E13.9, E14.0, E14.1, E14.6, E14.8, E14.9
Diabetes with chronic complication: E10.2 - E10.5, E10.7, E11.2 - E11.5, E11.7, E12.2 - E12.5, E12.7, E13.2 - E13.5, E13.7, E14.2 - E14.5, E14.7
Hemiplegia or paraplegia: G04.1, G11.4, G80.1, G80.2, G81.x, G82.x, G83.0 - G83.4, G83.9
Renal disease: I12.0, I13.1, N03.2 - N03.7, N05.2 - N05.7, N18.x, N19.x, N25.0, Z49.0 - Z49.2, Z94.0, Z99.2
Any malignancy, including lymphoma and leukemia, except malignant neoplasm of skin: C00.x - C26.x, C30.x - C34.x, C37.x - C41.x, C43.x, C45.x - C58.x, C60.x - C76.x, C81.x - C85.x, C88.x, C90.x - C97.x
Moderate or severe liver disease: I85.0, I85.9, I86.4, I98.2, K70.4, K71.1, K72.1, K72.9, K76.5, K76.6, K76.7
Metastatic solid tumour: C77.x - C80.x
AIDS/HIV: B20.x - B22.x, B24.x

“P5 I13 why only one year?”

P5 I25 Why was this limited to unplanned care? What is the justification for examining only a year’s previous admissions?”

Author reply: We focused specifically on GP contacts and unplanned admissions within the preceding year for two reasons. Primarily, this was because such encounters were judged to be close enough to the index admission to assume that all patients could be assumed to have an established history of excessive alcohol consumption (if only they had been screened effectively). Given the natural history of ARLD, we proposed that the absence of any relevant code in these contacts “within the year” of index emergency admission would imply missed detection of harmful alcohol-use. This blanket assumption of “missed opportunities” becomes less credible for healthcare contacts that occurred in the more distant past.

Secondly, with respect to GP contacts, a one-year period of prior CPRD practice registration is the minimum period for eligibility. Hence, not all CPRD patients have more than one-year of continuous general practice data recording before their “index” date. Variability in the duration of continuous CPRD practice registration is a recognised limitation of this dataset. We did not examine more distant GP records, which might cover varying periods of follow-up. As indicated above, our focus was on first emergency admission with acute problems related to liver disease.

“P6 I33 This is the only place you mention linkage to intensive care data. Please can you expand on this a little more - is this the Critical Care Minimum Dataset?”

Author reply: We had already stated this under data sources in the first paragraph of Methods, as follows. Hence, no changes. ACCMD was available from 2008, which covers our observation period.

“P6 I4-8 Please specify (perhaps in appendix) a) which codes were used and b) whether only underlying cause of death or any cause of death was used.”

Author reply: Manuscript updated. See revision and Supplementary Appendix 2 for further details.

6

“P7 I141-44 ‘these figures support the assertion..’ this is interpretive and belongs in discussion, not results.”

Author reply: Sentence removed.

“P7 I150-52 “This leaves almost one in four with no preceding record of an alcohol-related diagnosis (despite subsequently being readmitted as an emergency with ARLD)” I have a few concerns with this statement:

-The group with no preceding record of alcohol-related emergency admission in the last year are the 59% who have no emergency admission in the preceding year at all, plus the 24% \times 41% \approx 9% who had an emergency admission, so \sim 68% of the total.

-This isn’t the same as saying they have no preceding record of an alcohol-related diagnosis: what about primary care and non-emergency admissions? What about diagnoses more than a year before the index admission?

-In any case this kind of interpretive statement is again better in the discussion”

Author reply: We have updated and re-phrased to clarify. Here, we are referring specifically to “missed opportunities” during emergency admissions in the preceding year to detect and record liver disease (i.e. we are reporting the percentage of admissions without a relevant code recorded). So, as shown in Table 1, 76% (5,525 of 7,265) of emergency admissions in the preceding year had a alcohol-specific code listed as a discharge diagnosis. This leaves 24% (1,740 of 7,265) of such emergency admissions “lacking” a code. From this, we infer that the admitting teams did not screen, detect and record alcohol-related harms during that prior emergency admission. The legend to Table 1 indicates the correct the denominator. The expanded Methods section further emphasizes our focus is on “missed opportunities” among patients who had a contact in the year before admission. The denominator population for these figures are the people who were admitted as an emergency. Our purpose was to illustrate that when prior emergency hospitalisations occurred within just a year of admission for ARLD, an alcohol-related diagnosis was “missing” in 1 in 4 cases.

“P7 I158 onwards. Stage of liver disease – see comments above. It would be supportive for the analyses that follow to move the K-M plot in supplementary figure 1 D to the main body.”

Author reply: Yes, agreed, this deserved more emphasis. See our earlier comments relating to the importance of reporting data by “recorded-stage”. We have moved the relevant KM curve accordingly, incorporating it into Figure 2 in the revision (Fig 2H).

“P8 I114-21 This paragraph simply describes patterns in the figures without adding important information such as proportions and the results of statistical tests, which can only be found in the appendix table. Your statements “the proportion of first admissions with severe disease showed limited variation across deprivation quintiles” and “the proportion with “severe liver disease” (codes for cirrhosis or hepatic failure) was similar among men and women” seem to contradict the statistical testing reported in S1, which found a difference in the proportions significant at the $p=0.001$ level for both of these characteristics. Perhaps statistically significant but not large/important differences?”

Author reply: We had tried not to place too much emphasis on quite small (but statistically significant) differences. The relevant analyses and stats are in Supplementary Materials and we do not believe these deserve greater emphasis given the volume of other data presented.

“P8 I123-34. Perhaps with these small differences it would be useful to give age and percentages to one rather than zero decimal places in this paragraph and arguably could do so throughout paper.”

Author reply: Done

7

“P8 I134 P8 I134-40 ‘Collectively these findings...’ Interpretive and better in the discussion, not results. If changing coding practices over time is a genuine concern, would like to see more

in-depth discussion of this and how it impacts on interpretation in the discussion section".
Author reply: We have expanded the relevant part of the discussion. "Coding drift" is an ever-present possibility in all studies using HES data over long periods but most authors simply ignore or fail to report time-trends in covariates that might be vulnerable to evolving coding practice. The published literature using HES is replete with papers that apply the CCI score as a risk-adjuster without considering time-dependent variation in depth or completeness of discharge coding. Our sensitivity analyses excluding comorbidity entirely did not change the key conclusions in relation to time trends. The same was the case for models excluding recorded-stage (Liver Failure vs Not). See our revised commentary and expanded Supplementary Materials.

P8 1142 'Concerningly.' Emotive and should be replaced.

Author reply: Revised.

"P8 145. P=0.13 is really rather weak evidence for any change in the proportion of GP contacts with a record of alcohol, not sure it's valid to report it as a reduction. Separately, it would be useful to report the proportion of all patients who a GP contact with a record of alcohol/liver disease codes, i.e. with the denominator being all patients not just those with a GP contact in the last year. Otherwise, a decrease in the proportion with liver disease codes could for example be driven by an increase in proportion with a GP contact in last year, but no decrease in the overall recognition of liver disease"

Author reply: Re-phased and clarified. The lack of significant change is taken to demonstrate a lack of "improvement" over time (i.e. no significant increase). The focus here is on "missed opportunities" during primary care contacts within a year of index admission. We assume a history of excessive alcohol consumption would well-established at this time and elective diagnosis of underlying liver disease possible. Please see the extensive list of READ codes we used, which sought evidence of any hint of alcohol or liver problems being suspected during those prior contacts. As with the data on prior emergency admissions, the relevant denominator is the proportion of GP consulters – i.e. what proportion of those who were seen at the practice during the year before admission had any alcohol, or liver, codes recorded in those encounters.

"P9 157 How did you deal with patients transferred between hospitals when calculating length of stay?"

Author reply: We did not explicitly look at inter-hospital transfers. Length of stay is not a major focus of the paper. We don't believe that accounting for a minority of such transfers would have a significant impact on aggregated national-level trends.

"P9 117 when you refer to deaths within 48 hours of admission, does this mean within 1 day of admission or 2 days of admission? Note that a death within 2 days of admission does not necessarily mean within 48 hours, since neither time of admission nor death is available in these data, e.g. patient admitted 3am Monday and died 3pm Wednesday. Use of 'just 48 hours' emotive."

Author reply: Agreed. Clarified in revision.

P9 1145-47 Or died before transfer to ICU? Suggest move this to discussion section anyway.

Author reply: Revised.

"P10 15. I can't find a Table 3 in the main ms."

Author reply:

8

This was a typo – the text is referring to model results presented in Table 2.

"P10 1114-16 Again think an additional decimal place would be useful on the CFRs"

Author reply: Agreed. Done.

"P11 115-10. Please put results of sensitivity analyses in appendix"

Author reply: Agreed. Done.

Reviewer: 2

Dr. Cristal Brown, The University of Texas at Austin

Comments to the Author:

“Manuscript is well done and most significant consideration in revision is the use of stage given the high number of unspecified diagnoses. The model consolidates for those with known advanced liver disease and this may be an appropriate strategy throughout the manuscript.”

Author reply: We have addressed this similar point in response to Reviewer 1 (please see above). In short, the reporting of stage-specific data was intended to illustrate the limitations of ICD-10 coding and place the work in context of the existing literature and ‘routine statistics’ (which frequently focus on selective reporting of ‘stage specific’ cohorts, ignoring the ‘unspecified’ grouping). This is an important aspect of the work (at least for HES-based research from UK context). We report stage-specific data throughout to cover the entire cohort of cases identified using the LAA case definition. When it comes to the final risk-adjustment model, as stated by Reviewer 2, we dichotomize stage into coding for “Liver Failure” or “Not”, which is justified by the preceding data demonstrating KM curves by recorded-stage. See also the data on sensitivity analyses included in the revised Supporting Materials.

Peer Review - Outcomes of first emergency admissions for alcohol-related liver disease in England over a ten year period: Retrospective observational cohort study using linked electronic databases

1. First reading – big picture

a. Significance of the research question

i. Research question is significant. Alcohol-associated liver disease is prevalent, underdiagnosed and associated with high morbidity and mortality.

b. Originality of the research

i. This study uses nationwide, linked data and applies an innovative algorithm to more accurately identify patients with ARLD for characterization and reporting of mortality trends over time

c. Writing style and structure

i. Writing style is appropriate such that readers unfamiliar with the topic can follow the methodology, results and conclusions

ii. Ethical considerations are appropriately addressed.

d. Summarize the paper in a few sentences or a paragraph

i. This study uses a novel algorithm, previously tested and manually reviewed with consistent accuracy, to identify patients with ARLD presenting for an initial emergency admission. Using linked data, the authors characterize the population and provide case fatality with description of trends in mortality over 10 years spanning 5 years prior to and after the NCEPOD report which prompted attention to improving care for patients with ARLD. This study concludes that there are continued missed opportunities in primary and secondary care to identify these patients and ongoing high mortality rates in the first year following initial emergency admission.

2. Second reading

Look for possible improvements to clarify the presentation

Provide specific suggestions

a. Abstract

9

i. Adequately summarizes the findings of the study and provides concise and pertinent information.

b. Introduction

i. The authors do a good job of establishing the importance of the research topic.

There has been data driven concern for the underdiagnosis and inadequate care for patients with ARLD. Strong citations of the NCEPOD and Lancet reports noting unacceptably high levels of mortality in this hospitalized patient

population

- ii. These authors acknowledge the gap in the literature regarding characterization of this population on a nationwide level
- iii. They utilize an algorithm that they have previously tested and manually confirmed accuracy to improve the ability to detect patients with ARLD and link to available hospital administrative and death data.
- iv. Study aims are clearly defined.

c. Methods

- i. The research design is appropriate to address the study aims
- ii. Although there is sufficient description of the data sources, case definitions, and outcomes, I would recommend clarification of the following items:

1. Under “Population of interest”, provide an explanation for defining index admission as having no such admission within the preceding 10 years as opposed to no prior admissions at all.

Author reply: Thank you. This was based on the constraints of the available linked dataset. The current “release” of the HES Admitted Patient Care dataset linked to CPRD covers the period April 1997 – March 2021 inclusive. Hence, a ten-year “look-back” window of HES data was available for all cases in time period of interest (April 2008 – 31st March 2018). As described in our methods, our “first admissions” were admission-free for ten years. We accept that a minority of patients with ARLD might have had an admission in the much more distant past, which would imply an immediate sustained period of abstinence and recovery without needing any repeat emergency hospitalisations. Such cases could, in theory, present acutely for a second time over a decade later. This is a rare trajectory in our experience, given the high 1-year mortality (one in three) and very high readmission rates observed in patients with alcohol misuse disorder following index acute hospitalization for liver disease. But, we have placed greater emphasis on this limitation in the revised discussion.

2. Under “Patient-level covariates derived from the index admission”, why was hepatic encephalopathy not included as a co-variate for advanced liver disease? Ascites, variceal hemorrhage and HE are the 3 major complications associated with decompensated liver disease and the omission of this symptom should be addressed.

Author reply: Agreed – See updated discussion. The LAA algorithm includes a number of codes that are compatible with encephalopathy and are permitted to appear as a primary diagnosis (provided such codes are accompanied by an ARLD-specific code in the coding sequence). However, the use of the individual specific code for hepatic encephalopathy within HES is uncommon – just 2.5% of admissions in the present study, and <1% of cases lacking either varices or ascites codes. During our development of LAA, we concluded that reporting of the prevalence of encephalopathy on the basis of this single code simply lacked face validity. In our reviews of local case records we found that true encephalopathy was often coded using less specific codes for drowsiness or confusion. However, these same nonspecific codes could be recorded to denote alcohol intoxication or withdrawal. This problem

does not apply to the use of codes for ‘ascites’ or ‘varices’, where their recording is common and they can be taken as specific indicators of advanced liver disease when present in an admission for ARLD. The problem of capturing hepatic encephalopathy in ICD-10 is well documented and has prompted a recent update to codes available for this specific condition

(this occurred after our study time frame):

10

<https://www.ajmc.com/view/new-icd-10-code-aims-to-provide-more-insight-into-hepaticencephalopathy>

3. Under “Patient-level covariates derived from the index admission”, it would also be helpful to explain the decision to separate CCI categories of 0-1 and ≥ 2 . Is this standard or provide

an explanation for this choice

Author reply: The original CCI includes a number of ICD-10 codes for liver disease which would lead to over-estimating co-morbidity burden and/or 'double counting' of certain codes in risk adjustment (i.e. the original CCI includes various codes for 'mild' and 'moderate-to-severe' liver disease but many of the same codes are required for ARLD-specific cohort membership or condition-specific covariates). Our focus was to capture the contribution of "non-liver comorbidities". We derived a simple categorized variable based on non-liver components of the original index (listed above). It is routine to categorize the original CCI for risk-adjustment models (rather than using continuous scores). For binary logistic regression models it is standard practice to define simple categorical or binary variables for covariates (where possible). This aids the interpretation of odds ratios (as opposed to using continuous variables). Our non-liver comorbidity variable represents a simplification and adaptation of the original CCI and we acknowledge this. The models confirm that this variable was independently-associated with case fatality, along with recorded-stage and the additional 'flags' (ascites and varices). See also our sensitivity analyses (using absolute 'scores') in Supporting Materials.

iii. There are no concerns with the statistical methods utilized and their methodological description

Author Reply: Thank you. No additional changes required.

d. Results

i. The results are generally reported appropriately based on the stated objectives and methods.

ii. I would consider the following:

1. I have concerns regarding the use of stage of liver disease as a covariate and analysis of its association with mortality. Almost 40% of the patients have unspecified stage of liver disease. This means that the true percentage of patients presenting with early and advanced liver disease cannot be known. The authors do acknowledge the high rate of unspecified stage and include the supplemental table 1 to suggest a reasonable portion have advanced disease based on the presence of ascites and varices but I would caution using this variable for reporting association in the time trends in case fatality. The multivariate model uses only advanced stages and this is appropriate given the coding issue.

Author Reply: Please see our earlier response to Reviewer 1, where we explain why we believe it is vital to report data relating to "recorded-stage" of liver disease to place the current work, and existing literature, in context. Our research is intended to highlight the strengths and limitations of alternative approaches for interrogating such datasets. Published research and public health statistics derived from English Hospital Episode Statistics have frequently relied on stage-specific codes to derive cohorts and report burden or outcome. As recommended by Reviewer-2, we have moved the relevant KM survival curve (by recordedstage) to the main manuscript and emphasized the key learning points from this analysis.

Survival curves for patients who have a recorded-stage of liver disease (i.e. a stage-specific code) demonstrated the expected patterns (i.e. "worst" for liver failure, intermediate for

11

cirrhosis and "best" for hepatitis). However, we demonstrate that "unspecified" stage does not imply early or mild disease and that this group exhibits an intermediate prognosis reflecting heterogeneity in true stage. Hence, when proceeding to model time trends we dichotomized the stage into "Liver Failure" or "Not", relying on the additional covariates ("Ascites" and "Varices") to risk-adjust for other markers of advanced liver disease that are discrete from the recorded-stage derived from the liver code. Deleting the data relating to stage-specific coding would make it difficult to compare the current work with the previous literature and we aimed to illustrate the true significance of this often-ignored "unspecified" group.

2. In “Time trends in case-mix”, the last sentence of the first paragraph regarding the improvement in coding over time should be included in the discussion and not reported in results.

Author Reply: Agreed. Done.

3. There are multiple instances when a non-significant p-value is reported as having a significant change or trend toward significance. These should be corrected.

a. Concerningly, among patients with prior GP contact in the preceding year we found reductions in the proportion of consultations with a record of alcohol (46.6% vs. 48.1%, $p=0.13$)

b. There was also a small reduction in recording of alcohol-specific codes in prior emergency admissions (75.6% vs. 76.7%, $p=0.3$).

c. This did not quite reach significance for the 365 day end-point (33% vs 35%, $p=0.063$)

d. Overall crude rate reduced from 36.8% to 32.2% ($p=0.02$) and stage-specific crude rates showed downward trends – hepatic failure reducing from 59.2% to 53.9% ($p=0.07$), cirrhosis from 46.1% to 36.3% ($p=0.02$) and alcoholic hepatitis from 28.3% to 21.7% ($p=0.15$).

Author Reply: Agreed – we have revised the relevant sections.

4. Under “Deaths following first admission”, the following sentence belongs in discussion and not results. “These data provide further indirect evidence to support the validity of the LAA for cohort discovery and highlight the risk of missing a substantial mortality burden using HES-based statistics derived using the traditional primary definition.”

Author Reply: Agreed - moved

5. If stage-specific outcomes for time trends are retained, I recommend separating those stages with significant p values from those with insignificant p values

Author Reply: Agreed – revised

6. Need to add p value at the end of this statement. “Stage-specific reductions in odds of dying during first admission were 3% per year for hepatic failure ($p=0.004$), 6% for cirrhosis ($p<0.001$) and 7% for alcoholic hepatitis”.

Author Reply: Agreed – revised

e. Discussion

i. I would consider removing the first paragraph of the discussion. It is a restatement of the background which has been addressed in the introduction.

Author reply: Agreed – manuscript revised.

12

ii. Also consider changing the order of the paragraphs in the discussion. Readers would like a summary of the important results first – related to the primary objectives of the discussion, followed by the importance of these results. It takes 3 to 4 paragraphs before you address the major findings.

Author reply: Discussion has been revised.

f. Conclusion

i. I agree with the conclusions and limitations stated by the authors. The study successfully achieves its objectives in finding some improvement in prognosis for patients with ARLD but with ongoing concerns for late diagnosis and need for improved hospital care.

Author reply: Thank you, hence no additional changes made

g. Figures and Tables

i. The figures and tables in the manuscript and supplemental materials are pertinent to the findings and are consistent with the reported results. They are appropriately described and consistently reported throughout.

Author reply: Thank you, hence no additional changes made.

h. References

i. Relevant and up to date references provided

Author reply: Thank you, hence no additional changes made.

VERSION 2 – REVIEW

REVIEWER	Bernal, William King's College Hospital
REVIEW RETURNED	24-Oct-2023

GENERAL COMMENTS	We thank the authors for their comprehensive response to the review. Overall, our concerns have been satisfactorily addressed. A few minor comments remain which may be worth further consideration. It may be useful to make it explicit in the “Population of interest: First emergency admissions for ARLD” paragraph that an elective admission in the preceding ten years would not exclude inclusion of the patient. There are a number of statements in the ms which could still imply it is the first admission overall and might usefully be revised. The opening sentence of the discussion ‘Our study represents the first detailed description of characteristics and outcomes of index emergency admissions in the English population’ might better be expressed as ; ‘Our study represents the first detailed description of characteristics and outcomes of index emergency admissions with ARLD in the English population’
---

REVIEWER	Brown, Cristal The University of Texas at Austin, Internal Medicine
REVIEW RETURNED	27-Oct-2023

GENERAL COMMENTS	The authors have provided clear justifications and acknowledgement for each point of feedback. They have adequately addressed all items and updated the manuscript as appropriate. I have no concerns with moving forward with publication. The "Data Sources" paragraph has an error in April 1997 to Mark (March) 2021 that should be updated. Methods - Case definitions for ARLD admissions - change alternative to alternative
--

VERSION 2 – AUTHOR RESPONSE

Reviewer: 1

A few minor comments remain which may be worth further consideration.

1. "It may be useful to make it explicit in the "Population of interest: First emergency admissions for ARLD" paragraph that an elective admission in the preceding ten years would not exclude inclusion of the patient. There are a number of statements in the ms which could still imply it is the first admission overall and might usefully be revised"

Author Response: Agreed, an additional sentence has been added to this paragraph to state this explicitly. We have also adjusted a number of other sentences by adding 'emergency' prior to 'admission' to further emphasize this point.

2. The opening sentence of the discussion 'Our study represents the first detailed description of characteristics and outcomes of index emergency admissions in the English population' might better be expressed as ; 'Our study represents the first detailed description of characteristics and outcomes of index emergency admissions with ARLD in the English population'

Author Response: Agreed, the relevant opening sentence has been adjusted.

Reviewer: 2

The authors have provided clear justifications and acknowledgement for each point of feedback. They have adequately addressed all items and updated the manuscript as appropriate. I have no concerns with moving forward with publication.

The "Data Sources" paragraph has an error in April 1997 to Mark (March) 2021 that should be updated.

Authori Response: Agreed, typo corrected

Methods - Case definitions for ARLD admissions - change alterntive to alternative

Author Response: Agreed, typo corrected